# MEMORY FORCING: SPATIO-TEMPORAL MEMORY FOR CONSISTENT SCENE GENERATION ON MINECRAFT

## ABSTRACT

Autoregressive video diffusion models have proved effective for world modeling and interactive scene generation, with Minecraft gameplay as a representative application. To faithfully simulate play, a model must generate natural content while exploring new scenes and preserve spatial consistency when revisiting explored areas. Under limited computation budgets, it must compress and exploit historical cues within a finite context window, which exposes a trade-off: Temporal-only memory lacks long-term spatial consistency, whereas adding spatial memory strengthens consistency but may degrade new scene generation quality when the model over-relies on insufficient spatial context. We present *Memory Forcing*, a learning framework that pairs training protocols with a geometry-indexed spatial memory. *Hybrid Training* exposes distinct gameplay regimes, guiding the model to rely on temporal memory during exploration and incorporate spatial memory for revisits. *Chained Forward Training* extends autoregressive training with model rollouts, where chained predictions create larger pose variations and encourage reliance on spatial memory for maintaining consistency. *Point-to-Frame Retrieval* efficiently retrieves history by mapping currently visible points to their source frames, while *Incremental 3D Reconstruction* maintains and updates an explicit 3D cache. Extensive experiments demonstrate that Memory Forcing achieves superior long-term spatial consistency and generative quality across diverse environments, while maintaining computational efficiency for extended sequences.

## 1 INTRODUCTION

Autoregressive video models (Bar et al., 2025; Chen et al., 2024; Song et al., 2025) based on diffusion (Ho et al., 2020; Dhariwal & Nichol, 2021; Peebles & Xie, 2023) have recently emerged as powerful tools for world modeling, showing strong capabilities in interactive scene generation (Feng et al., 2024; Parker-Holder et al., 2024), particularly in open-world environments like Minecraft, where multi-dimensional controls enable rich user interactions. These models (Decart et al., 2024; Guo et al., 2025; Cheng et al., 2025) learn to predict future frames conditioned on past observations and user actions, enabling autoregressive (AR) rollouts that react to player inputs in real time. Within the AR paradigm, the model must condition on a context window of past frames, but latency and memory limits bound the window size. Therefore, it is critical to compress and prioritize historical information (*i.e.*, memory) within this limited window.

In prior works, the allocation of memory manifests in two characteristic failure modes, as shown in Fig. 1. Models that incorporate long-term spatial memory preserve consistency on revisits (Fig. 1(a)) but fail in novel scenes exploration. Conversely, temporal-only models fail to maintain spatial consistency upon revisit (Fig. 1(b)). Moreover, teacher-forced training Huang et al. (2025) underestimates inference-time drift, encouraging over-reliance on short-horizon temporal cues and underuse of retrieved memory at test time. These observations motivate a training framework that enables the model to modulate its reliance on temporal and spatial memory across exploration and revisit regimes, thereby balancing exploration flexibility and revisit consistency.

To address these limitations, we introduce Memory Forcing, a training framework that forces the model to flexibly and effectively use temporal and spatial memory under a fixed window. Specifi-

Figure 1: Two paradigms of autoregressive video models and their fail cases. (a) Long-term spatial memory models maintain consistency when revisiting areas yet deteriorate in new environments. (b) Temporal memory models excel in new scenes yet lack spatial consistency when revisiting areas.

cally, Hybrid Training uses distinct data distributions to emulate complementary gameplay regimes, so the model learns to rely on temporal memory for novel-scene exploration and to incorporate spatial memory on revisits for consistency. In practice, we adopt temporal-only conditioning on VPT (Baker et al., 2022) (human play, exploration-oriented) and spatial&temporal conditioning on MineDojo (Fan et al., 2022) (simulated trajectories with frequent revisits and adjacent viewpoints), achieving a balanced optimum across the two tasks. Besides, we introduce Chained Forward Training to augment autoregressive learning with model rollouts: it progressively replaces ground-truth temporal context with the model's own predictions, amplifies pose/viewpoint drift across windows, and thus encourages reliance on spatial memory to maintain revisit consistency.

Beyond the training protocol, we equip the model with Geometry-indexed Spatial Memory. Prior frame-level retrieval (Xiao et al., 2025; Chen et al., 2025) is appearance-based, sensitive to viewpoint and illumination changes, and prone to accumulating near-duplicate views under neighboring poses. As sequences grow, redundancy and lookup latency grow with the size of the memory bank. State-space methods (Po et al., 2025) compress history into latent states and alleviate this efficiency issue, but they lack explicit spatial indexing, making it difficult to specify which spatial evidence to retain and which redundancy to discard. Instead, we maintain a coarse scene representation via streaming 3D reconstruction and retrieve history with point-to-frame mapping: currently visible 3D points are back-traced to their source frames to select a compact, pose-relevant set of views. This geometry-anchored access is robust to viewpoint changes, bounds the candidate set (top-$k$), and scales with visible spatial coverage rather than sequence length.

We conduct comprehensive experiments on Minecraft benchmark (Fan et al., 2022) across three critical dimensions: long-term memory with spatial revisitations, generalization on unseen terrains, and generation in new environments. Our method achieves superior performance across all three settings compared to both temporal-only and spatial memory baselines, while our Geometry-indexed Spatial Memory demonstrates 7.3× faster retrieval speed with 98.2% less memory storage. These results demonstrate that Memory Forcing effectively resolves the trade-off between spatial consistency and generative quality while maintaining computational efficiency.

In summary, our contributions are threefold:

- We introduce Memory Forcing, a framework that simultaneously addresses capability trade-offs and efficiency limitations in memory-augmented video generation.

- We develop the Hybrid Training and Chained Forward Training strategy that teaches models to use temporal memory for exploration and incorporate spatial memory for revisits, and a Geometry-indexed Spatial Memory built via streaming 3D reconstruction with Point-to-Frame Retrieval, whose lookup cost scales with visible spatial coverage rather than sequence length.

- Extensive experiments demonstrate that Memory Forcing achieves superior performance in both spatial consistency and generative quality in new environments, while maintaining computational efficiency for extended sequences.

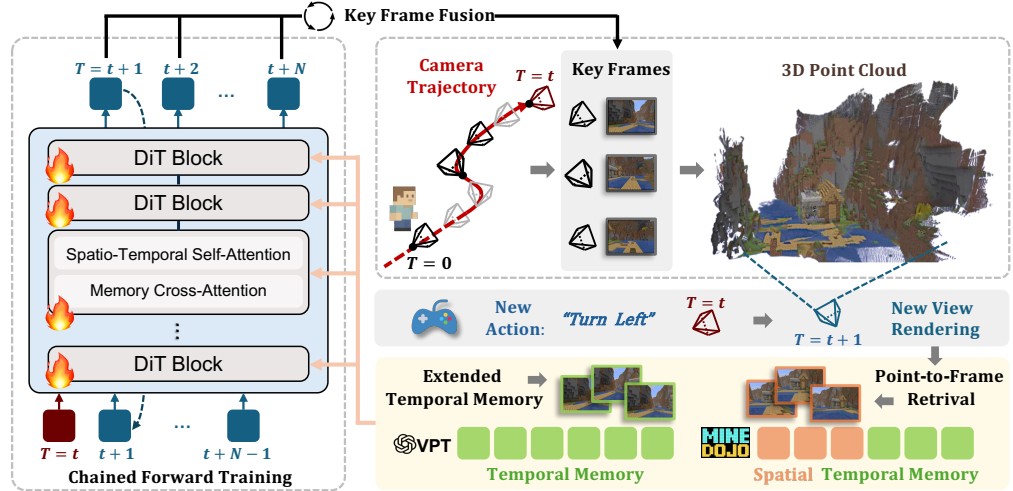

Figure 2: Memory Forcing Pipeline. Our framework combines spatial and temporal memory for video generation. 3D geometry is maintained through streaming reconstruction of key frames along the camera trajectory. During generation, Point-to-Frame Retrieval maps spatial context to historical frames, which are integrated with temporal memory and injected together via memory cross-attention in the DiT backbone. Chained Forward Training creates larger pose variations, encouraging the model to effectively utilize spatial memory for maintaining long-term geometric consistency.

## 2 RELATED WORKS

**Autoregressive Video Models.** Autoregressive video generation (Harvey et al., 2022; Li et al., 2025b; Xie et al., 2025; Wu et al., 2024; Teng et al., 2025; Henschel et al., 2025) enables long video synthesis by conditioning on preceding frames. Early token-based approaches (Wu et al., 2024; Kondratyuk et al., 2023) achieved temporal consistency but compromised visual fidelity. Recent diffusion-based methods (Voleti et al., 2022; Hong et al., 2024; Chen et al., 2024; Song et al., 2025) achieve superior quality through masked conditioning and per-frame noise control.

**Interactive Game World Model.** World models predict future states from current states and actions (Ha & Schmidhuber, 2018a;b; Hafner et al., 2019; 2020). Recent video generation advances have enabled interactive world models (OpenAI, 2024; Feng et al., 2024; Parker-Holder et al., 2024; Valevski et al., 2024; Zhang et al., 2025; He et al., 2025; Yu et al., 2025b; Che et al., 2024) for complex gaming environments. Minecraft's rich action space and environmental dynamics have inspired numerous game world models (Decart et al., 2024; Guo et al., 2025; Cheng et al., 2025; Po et al., 2025; Chen et al., 2025; Xiao et al., 2025). While models like MineWorld (Guo et al., 2025) and NFD (Cheng et al., 2025) show strong interactive capabilities, they lack long-term memory. State-space approaches (Po et al., 2025) introduce memory mechanisms but remain limited by training context length. WorldMem (Xiao et al., 2025) uses pose-based retrieval for long-term memory but suffers from limited novel scene generation and lacks real-time interactivity.

**3D Reconstruction and Memory Retrieval.** Learning-based 3D reconstruction was pioneered by DUSt3R (Wang et al., 2024), with subsequent multi-view extensions (Wang et al., 2025a;c; Yang et al., 2025) and streaming methods (Wang et al., 2025b; Wu et al., 2025) for sequential processing. SLAM-based approaches like VGGT-SLAM (Maggio et al., 2025) handle long sequences through incremental submap alignment. For memory retrieval in video generation, existing approaches range from pose-based methods (Xiao et al., 2025; Yu et al., 2025a) using field-of-view overlap to 3D geometry-based approaches like VMem (Li et al., 2025a) with surfel-indexed view selection.

## 3 MEMORY FORCING

We introduce Memory Forcing, a learning framework that pairs training protocols with geometry-indexed spatial memory to enable long-term spatial consistency. Our approach addresses the funda-

mental trade-off between temporal memory for generation and spatial memory for revisits through Hybrid Training and Chained Forward Training (CFT). Section 3.1 provides background on autoregressive video diffusion models and interactive game world modeling. Section 3.2 presents our memory-augmented model architecture. Section 3.3 details our Memory Forcing training protocols. Section 3.4 introduces our explicit 3D memory maintenance and retrieval approach.

## 3.1 PRELIMINARIES

**Autoregressive Video Diffusion Models.** Autoregressive Video Diffusion Models generate video sequences by predicting future frames conditioned on past observations. Following Diffusion Forcing (Chen et al., 2024), we denote a video sequence as $X^{1:T} = x_1, x_2, \ldots, x_T$ where each frame $x_t$ is assigned an independent noise level $k_t \in [0, 1]$. The model learns to predict noise $\epsilon_\theta(\tilde{X}^{1:T}, k^{1:T})$ where $\tilde{X}^{1:T}$ represents the noisy sequence and $k^{1:T} = k_1, k_2, \ldots, k_T$ are the noise levels. The training objective minimizes:

$$\mathcal{L} = \mathbb{E}_{k^{1:T}, X^{1:T}, \epsilon^{1:T}} \left[ |\epsilon^{1:T} - \epsilon_\theta(\tilde{X}^{1:T}, k^{1:T})|^2 \right] \tag{1}$$

This framework enables flexible conditioning patterns for autoregressive generation by allowing arbitrary combinations of clean and noisy frames within a sequence.

**Interactive Game World Model.** Interactive game environments present unique challenges for video generation models. Players navigate complex 3D environments where actions $A^{1:T}$ include movement commands, camera rotations, and object interactions that directly influence both immediate visual changes and long-term environment state evolution. For action-conditioned generation, the model predicts noise conditioned on both visual observations and actions: $\epsilon_\theta(\tilde{X}^{1:T}, k^{1:T}, A^{1:T})$ enabling the model to generate coherent video sequences that respond appropriately to player inputs.

## 3.2 MEMORY-AUGMENTED ARCHITECTURE

We follow previous works (Cheng et al., 2025) in adopting block-wise causal attention for efficient spatio-temporal modeling, adaLN-zero conditioning for action integration, and 3D positional embeddings within a Diffusion Transformer (DiT) backbone. To incorporate long-term spatial memory into the generation process, we introduce several memory-specific components.

**Spatial Memory Extraction.** We employ the VGGT (Wang et al., 2025a) network with our cross-window scale alignment to enable streaming reconstruction. Historical frames are then efficiently extracted through Point-to-Frame Retrieval, providing accurate access to long-term spatial memory.

**Memory Cross-Attention.** We integrate Cross-Attention modules within each DiT block to leverage long-term spatial memory during generation. Retrieved historical frames serve as keys and values, while current frame tokens act as queries:

$$\text{Attention}(\tilde{Q}, \tilde{K}_{\text{spatial}}, V_{\text{spatial}}) = \text{Softmax}\left(\frac{\tilde{Q}\tilde{K}_{\text{spatial}}^T}{\sqrt{d}}\right) V_{\text{spatial}} \tag{2}$$

where $\tilde{Q}$ and $\tilde{K}_{\text{spatial}}$ are queries and keys augmented with Plücker coordinates to encode relative pose information between current and historical viewpoints.

## 3.3 AUTOREGRESSIVE DIFFUSION TRAINING WITH MEMORY FORCING

Memory-augmented video generation models face a fundamental capability trade-off. Models relying heavily on long-term spatial memory generate content consistent with previously visited scenes, but degrade when generating new scenes due to insufficient relevant spatial memory. We therefore propose Memory Forcing training protocols that teaches models to dynamically balance these two capabilities, learning when to rely on temporal context versus spatial memory.

**Hybrid Training.** Our hybrid training approach operates within a fixed context window of $L$ frames. We strategically allocate half the window ($L/2$ frames) as fixed temporal context frames, while the remaining $L/2$ frames are flexibly assigned based on the scene context. The complete context

window construction can be formalized as:

$$\mathcal{W} = [\mathcal{T}_{\text{fixed}}, \mathcal{M}_{\text{context}}] = \begin{cases} [\mathcal{T}_{\text{fixed}}, \mathcal{M}_{\text{spatial}}] & \text{if revisiting previously observed areas} \\ [\mathcal{T}_{\text{fixed}}, \mathcal{T}_{\text{extended}}] & \text{if exploring new scenes} \end{cases} \tag{3}$$

where $\mathcal{T}_{\text{fixed}}$ represents the fixed $L/2$ recent temporal context frames, $\mathcal{M}_{\text{spatial}}$ represents long-term spatial memory retrieved by our Geometry-indexed Spatial Memory 3.4, and $\mathcal{T}_{\text{extended}}$ represents additional temporal frames from earlier time steps. This dynamic allocation enables the model to leverage the most appropriate memory source for each generation scenario.

Inspired by Figure 1, we apply different memory strategies to different datasets: spatial memory for synthetic dataset (Fan et al., 2022) with frequent area revisiting, and extended temporal context for VPT dataset (Baker et al., 2022) with new scene generation.

**Chained Forward Training.** We introduce Chained Forward Training (CFT) to enhance our hybrid training strategy. CFT sequentially processes temporal windows where predicted frames from earlier windows are incorporated into subsequent windows, creating cascading dependencies across the sequence. Details are shown in Algorithm 1 in the Appendix. At each step $j$, the temporal window $\mathcal{W}_j$ contains both ground-truth frames $\mathbf{x}_k$ and previously predicted frames $\hat{\mathbf{x}}_k$, leading to the loss:

$$\mathcal{L}_{\text{chain}} = \frac{1}{T} \sum_{j=0}^{T-1} \mathbb{E}_{t,\epsilon} \left[ \| \epsilon - \epsilon_\theta(\mathcal{W}_j(\mathbf{x}, \hat{\mathbf{x}}), \mathcal{C}_j, t) \|^2 \right], \quad t \sim \text{Uniform}(0, T_{\text{noise}}), \epsilon \sim \mathcal{N}(0, \mathbf{I}) \tag{4}$$

This approach extends autoregressive training with model rollouts, where larger pose variations created by chained predictions cause inaccuracies to propagate from earlier windows, encouraging the model to rely on spatial memory for maintaining consistency across revisited areas. Additionally, by replacing ground truth temporal context with the model's own predictions during training, this approach helps reduce accumulation errors that typically arise during autoregressive inference.

### 3.4 GEOMETRY-INDEXED SPATIAL MEMORY

Our Geometry-indexed Spatial Memory maintains explicit scene geometry and enables efficient retrieval of long-term historical visual information based on 3D spatial relationships. This approach consists of two key components: Point-to-Frame Retrieval for identifying relevant historical frames and Incremental 3D Reconstruction for maintaining and updating scene representations.

**Point-to-Frame Retrieval.** For each current frame, we project the global point cloud to the current camera pose and analyze the source frame indices of visible points to identify the most relevant historical frames:

$$\mathcal{H}_t = \arg \max_{k=1,\ldots,8} \text{Count}(\text{source}(p_i) : p_i \in \mathcal{P}_{\text{visible}}^t) \tag{5}$$

where $\mathcal{P}_{\text{visible}}^t$ represents the set of points visible under the current camera pose for frame $t$, $\text{source}(p_i)$ denotes the source frame index of point $p_i$, and $\mathcal{H}_t$ contains the top-8 most frequently referenced historical frames among the visible points. This retrieval mechanism maintains $O(1)$ complexity regardless of sequence length, enabling scalable processing.

**Incremental 3D Reconstruction.** We adopt a selective reconstruction approach that dynamically determines keyframes based on spatial information content. A frame qualifies as a keyframe when it either reveals previously unobserved regions or when insufficient historical context exists:

$$\text{IsKeyframe}(t) = \mathcal{C}(I_t^{\text{proj}}) \vee (|\mathcal{H}_t| < \tau_{\text{hist}}) \tag{6}$$

where $\mathcal{C}(I_t^{\text{proj}})$ determines whether the current view contributes new spatial coverage when projected onto existing geometry, and $\tau_{\text{hist}} = 8$ serves as the minimum historical frame count threshold.

Upon reaching window capacity, we jointly process keyframes, historical frames selected via Point-to-Frame Retrieval for improved geometric consistency, and overlapping frames from the previous window that provide depth scale reference for aligning the new window. VGGT generates relative depth maps and confidence scores for each frame in this window, followed by our cross-window scale alignment module that establishes consistent depth scale across windows through correspondence analysis in overlapping regions. 3D geometry is reconstructed through depth map back-projection using camera extrinsics derived from quaternion-composed poses:

$$\mathbf{E} = \begin{bmatrix} \mathbf{R}(pitch, yaw) & -\mathbf{R}\mathbf{C} \\ \mathbf{0}^T & 1 \end{bmatrix} \tag{7}$$

where $\mathbf{R}(pitch, yaw)$ encodes the viewing orientation through quaternion-based rotation composition, and $\mathbf{C} = [x, y, z]^T$ specifies the camera's spatial position. The reconstructed geometry is subsequently integrated into our global representation through spatially-aware voxel sampling.

This design achieves efficient scene representation and retrieval through two key mechanisms. First, selective keyframe reconstruction processes and stores only frames that contribute new spatial coverage, preventing redundant computation and storage when revisiting encountered areas. Second, voxel downsampling maintains an upper bound on point density for any pose region, ensuring constant retrieval complexity regardless of temporal sequence length or scene scope. These mechanisms collectively ensure that memory consumption scales with spatial coverage rather than temporal duration, enabling efficient processing of extended sequences.

## 4 EXPERIMENTS

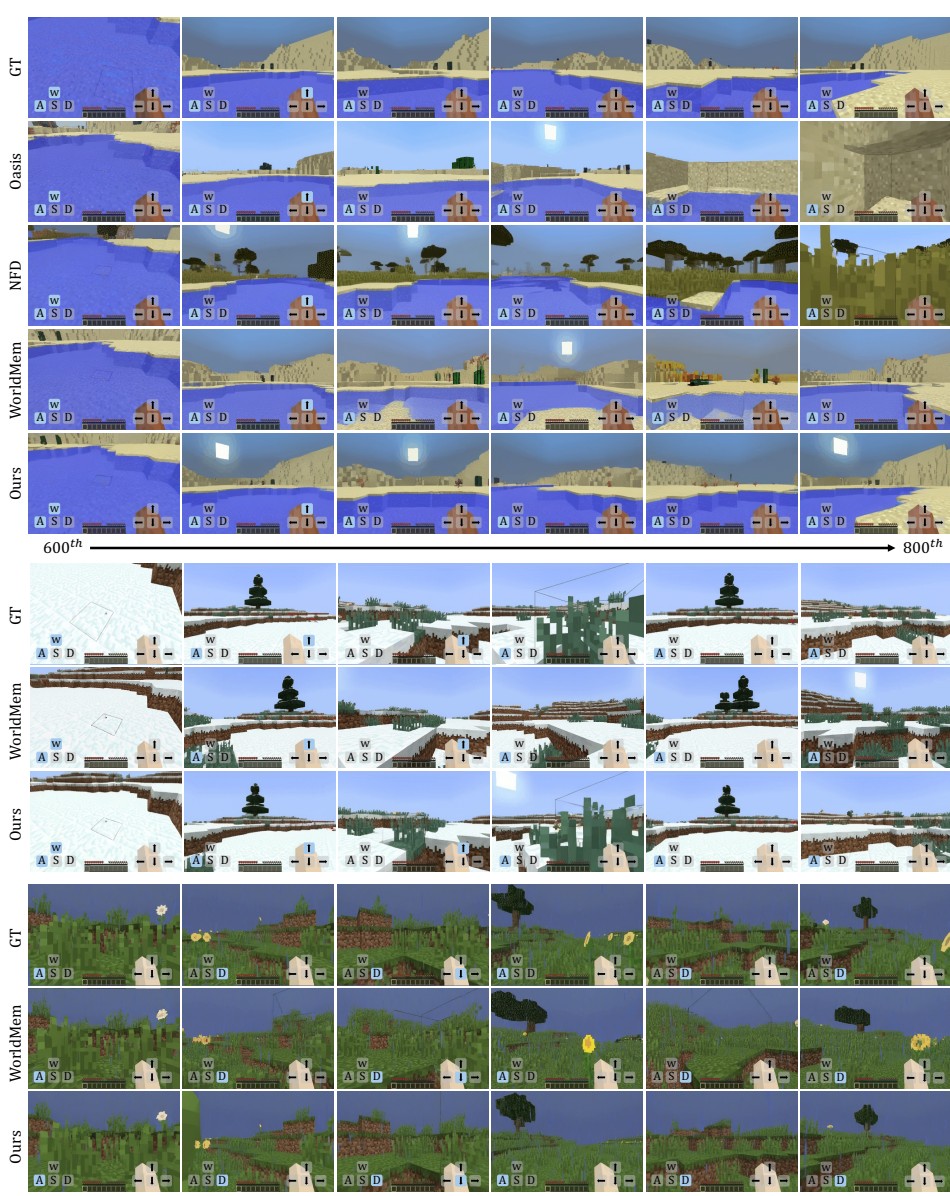

Figure 3: Memory capability comparison across different models for maintaining spatial consistency and scene coherence when revisiting previously observed areas.

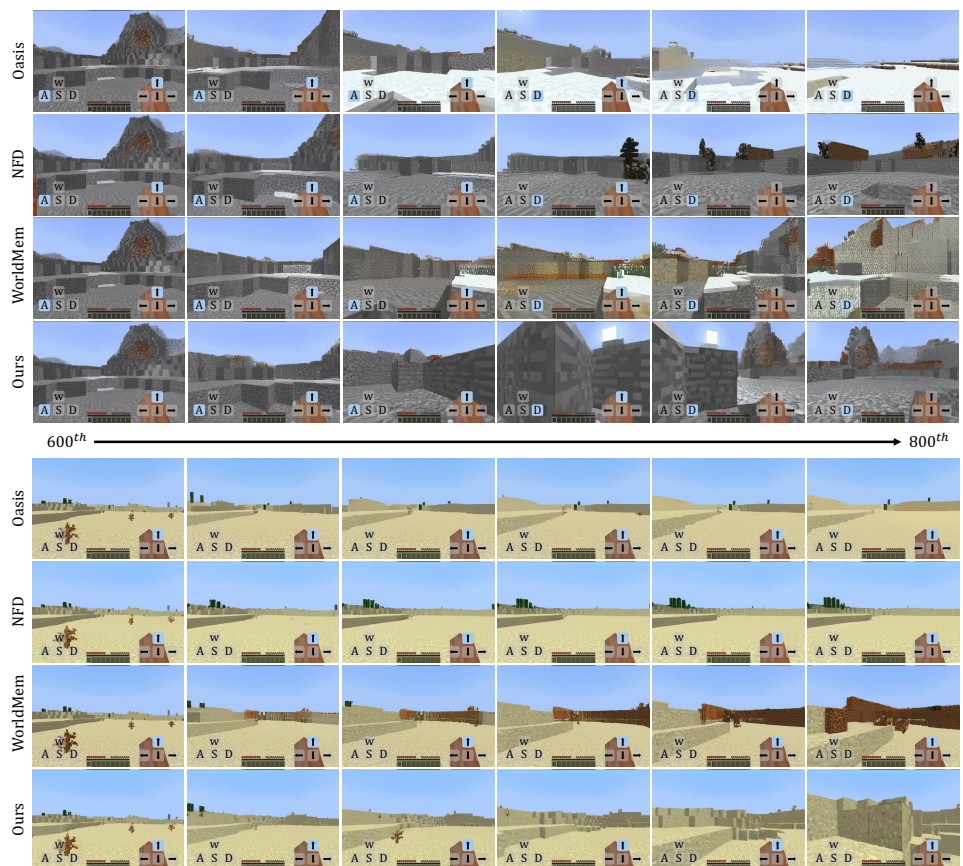

Figure 4: Generalization performance on unseen terrain types (top) and generation performance in new environments without historical spatial memory (bottom) across different models.

We conduct comprehensive experiments to evaluate our Memory Forcing framework through both quantitative and qualitative analyses. We demonstrate our model's long-term memory capabilities, generalization, and generation performance in new environments on our constructed Minecraft benchmark, and assess the retrieval and storage efficiency of our Geometry-indexed Spatial Memory. Additionally, we provide ablation studies on our frame retrieval strategy and training methodology.

## 4.1 EXPERIMENTAL SETUP

**Implementation Details.** Our model converges after approximately 400k training steps across 24 GPUs with batch size of 4. We employ the Adam optimizer with a learning rate of 4e-5. All training and evaluation are conducted on NVIDIA H20/H100 GPUs using PyTorch. We employ a 2D variational autoencoder following NFD (Cheng et al., 2025) for frame tokenization, providing 16× spatial compression and transforming each frame into $24 \times 14$ continuous tokens. Video frames are resized to $384 \times 224$ resolution, maintaining the original aspect ratio and sufficient visual detail.

**Baselines.** We compare our approach against baseline models including Oasis (Decart et al., 2024) and NFD (Cheng et al., 2025), as well as the long-term memory model WorldMem (Xiao et al., 2025). For fair comparison, all models use a 16-frame context window during both training and evaluation. All models follow their respective training configurations and are trained on identical Synthetic datasets for approximately 500-600k steps to ensure consistent evaluation conditions.

**Datasets.** For training, we use the VPT (Baker et al., 2022) dataset, which pairs 25-dimensional action vectors with corresponding video sequences. Following previous work (Guo et al., 2025), we exclude frames without actions or when the graphical user interface is visible to reduce noise. Additionally, we utilize a synthetic dataset generated from MineDoJo (Fan et al., 2022) for long-term memory training, following the configuration of WorldMem (Xiao et al., 2025), which consists

| Method | Long-term Memory | | | | Generalization Performance | | | | Generation Performance | | | |
|---|---|---|---|---|---|---|---|---|---|---|---|---|
| | FVD↓ | PSNR↑ | SSIM↑ | LPIPS↓ | FVD↓ | PSNR↑ | SSIM↑ | LPIPS↓ | FVD↓ | PSNR↑ | SSIM↑ | LPIPS↓ |
| Oasis | 196.8 | 16.83 | 0.5654 | 0.3791 | 477.3 | 14.74 | 0.5175 | 0.5122 | 285.7 | 14.51 | 0.5063 | 0.4704 |
| NFD | 220.8 | 16.35 | 0.5819 | 0.3891 | 442.6 | 15.49 | 0.5564 | 0.4638 | 349.6 | 14.64 | 0.5417 | 0.4343 |
| WorldMem | 122.2 | 19.32 | 0.5983 | 0.2769 | 328.3 | 16.23 | 0.5178 | 0.4336 | 290.8 | 14.71 | 0.4906 | 0.4531 |
| Ours | **84.9** | **21.41** | **0.6692** | **0.2156** | **253.7** | **19.86** | **0.6341** | **0.2896** | **185.9** | **17.99** | **0.6155** | **0.3031** |

Table 1: A comparison of different methods across various capabilities and evaluation metrics.

| Frame Range | 0–999 | | 1000–1999 | | 2000–2999 | | 3000–3999 | | Total (0–3999) | |
|---|---|---|---|---|---|---|---|---|---|---|
| Method | Speed (FPS↑) | Mem. (Count↓) | Speed (FPS↑) | Mem. (Count↓) | Speed (FPS↑) | Mem. (Count↓) | Speed (FPS↑) | Mem. (Count↓) | Speed (FPS↑) | Mem. (Count↓) |
| WorldMem | 10.11 | +1000 | 3.43 | +1000 | 2.06 | +1000 | 1.47 | +1000 | 4.27 | 4000 |
| Ours | **18.57** | **+25.45** | **27.08** | **+19.70** | **41.36** | **+14.55** | **37.84** | **+12.95** | **31.21** | **72.65** |

Table 2: Comparison of retrieval efficiency between WorldMem and our Geometry-indexed Spatial Memory across different sequence lengths. "Mem." denotes the number of frames in memory bank.

of 11k videos containing 1,500-frame action sequences with frequent pose transitions to previously visited spatial locations. For evaluation, we constructed three datasets using MineDojo to assess the model's performance across various aspects:

- **Long-term Memory**: 150 long video sequences (1,500 frames) were isolated from the World-Mem dataset (Xiao et al., 2025) to evaluate the model's capacity for long-term memory retention.

- **Generalization Performance**: We constructed 150 video sequences (800 frames) from nine unseen Minecraft terrains using MineDojo (Fan et al., 2022) to evaluate the model's generalization.

- **Generation Performance**: We constructed 300 video sequences (800 frames) using Mine-Dojo (Fan et al., 2022) to assess generation performance in new environments.

**Evaluation Metrics.** We evaluate our model's performance using established video quality metrics. We measure perceptual quality with Fréchet Video Distance (FVD) and Learned Perceptual Image Patch Similarity (LPIPS), while assessing pixel-level accuracy through Peak Signal-to-Noise Ratio (PSNR) and Structural Similarity Index Measure (SSIM). These metrics collectively provide comprehensive assessment of both visual fidelity and consistency in generated sequences.

### 4.2 MODEL CAPABILITIES ASSESSMENT

For all quantitative and qualitative evaluations, we generate frames 600-800 (200 frames) to assess long-sequence generation capabilities using the datasets described in our experimental setup.

**Long-term Memory.** We evaluate models' ability to maintain spatial consistency and scene coherence when revisiting previously observed areas using our long-term memory evaluation dataset. As demonstrated in Table 1, our method achieves superior performance across all metrics, indicating enhanced visual fidelity in long-sequence generation. Figure 3 further shows that our model demonstrates the most precise memory when returning to previously visited locations. While WorldMem exhibits some memory retention capabilities, it produces inaccurate and unstable view generation with visual artifacts in the generated scenes (e.g., the fifth frame in the fourth row). In contrast, the remaining baseline models lack long-term memory mechanisms, resulting in spatial inconsistencies where camera viewpoint changes inappropriately alter scene geometry and terrain features.

**Generalization Performance.** Using our generalization evaluation dataset with nine novel terrain scenarios, our approach demonstrates robust generalization performance, significantly outperforming baselines across all metrics as shown in Table 1, indicating strong adaptability to unseen environments. The top portion of Figure 4 illustrates qualitative generalization results, where our model generates stable and consistent outputs across novel terrains. In contrast, WorldMem and NFD exhibit artifacts in their generations, while Oasis shows scene inconsistencies.

**Generation Performance.** Our comprehensive evaluation using the generation performance dataset demonstrates that our method outperforms all baselines across metrics in Table 1, highlighting the effectiveness of balancing long-term and temporal memory. The bottom portion of Figure 4 illustrates generation performance in new environments, where our model exhibits responsive movement dynamics with distant scenes progressively becoming clearer as the agent approaches. In contrast,

| Training Strategies | | | Retrieval Strategies | | Metrics | | | |
|---|---|---|---|---|---|---|---|---|
| FT | HT-w/o-CFT | MF | Pose-based | 3D-based | FVD ↓ | PSNR ↑ | SSIM ↑ | LPIPS ↓ |
| ✓ | | | | ✓ | 366.1 | 15.09 | 0.5649 | 0.4122 |
| | ✓ | | | ✓ | 230.4 | 16.24 | 0.5789 | 0.3598 |
| | | ✓ | ✓ | | 225.9 | 16.24 | 0.5945 | 0.3722 |
| | | ✓ | | ✓ | **165.9** | **18.17** | **0.6222** | **0.2876** |

Table 3: Ablation study comparing training strategies and retrieval mechanisms. FT: full-parameter fine-tuning, HT-w/o-CFT: hybrid training without CFT, MF: Memory Forcing with HT and CFT.

WorldMem experiences significant quality degradation in this scenario, NFD shows minimal variation in distant scenes regardless of agent movement, and Oasis generates oversimplified distant scenes that lack proper distance-based visual transitions.

## 4.3 Efficiency of Geometry-indexed Spatial Memory

Table 2 evaluates the computational efficiency and storage requirements of our Geometry-indexed Spatial Memory compared to WorldMem's retrieval approach across 20 4000-frame MineDojo videos. While WorldMem stores all historical frames and performs linear-complexity retrieval across the entire collection, our selective keyframe approach reduces memory bank size by 98.2% while achieving 7.3× faster retrieval speed at 0-3999 frames. Efficiency gains increase with sequence length, reaching 25.7× speedup in the 3000-3999 frame range as WorldMem becomes increasingly slower. WorldMem's memory bank grows linearly with sequence length, while our Geometry-indexed Spatial Memory scales with spatial coverage expansion, storing only keyframes with new geometric information. Our speeds include the complete 3D memory pipeline (reconstruction and retrieval), while WorldMem's include pose-based retrieval across all stored frames.

## 4.4 Ablation Studies

Table 3 shows ablation studies on 300 videos from Long-term Memory and Generation Performance datasets analyzing the contributions of our training strategies and retrieval mechanisms.

**Training Strategy Analysis.** We compare three training approaches: full-parameter Fine-Tuning (FT) after VPT pre-training, Hybrid Training without Chained Forward Training (HT-w/o-CFT), and our complete Memory Forcing training strategy (MF). Direct fine-tuning achieves limited performance as the model struggles to balance temporal memory and spatial memory, typically over-relying on one modality at the expense of the other. HT-w/o-CFT demonstrates improvement by integrating real and synthetic data, but inadequately trains the model's dependence on spatial memory during spatial revisitation scenarios. Our Memory Forcing training approach achieves optimal performance by enabling the model to adaptively utilize temporal context when exploring new scenes while leveraging spatial memory when revisiting previously observed areas, effectively resolving the fundamental capability trade-off between generation quality and long-term memory retention.

**Retrieval Mechanism Comparison.** Our 3D-based approach substantially outperforms pose-based retrieval by leveraging explicit geometric representations for more precise identification of spatially relevant historical frames, while achieving superior computational efficiency as shown in Table 2.

## 5 Conclusions

We introduced Memory Forcing, a novel framework that addresses the fundamental trade-off between long-term spatial memory and new scene generation in autoregressive video models. Our approach consists of two key innovations: an efficient Geometry-indexed Spatial Memory that achieves constant-time retrieval complexity through streaming 3D reconstruction and point-to-frame retrieval, and a hybrid training strategy featuring Chained Forward Training that teaches models to dynamically balance temporal and spatial memory utilization. Our framework encourages adaptive contextual selection, relying on temporal memory for new scene generation while leveraging spatial memory for consistency in previously encountered areas. Extensive experiments demonstrate that Memory Forcing achieves superior performance in both spatial consistency and generative quality while maintaining computational efficiency for extended sequences, effectively resolving the capability trade-off that has limited prior memory-augmented video models.

## 6 ETHICS STATEMENT

We confirm that this research adheres to the ICLR Code of Ethics. We have carefully considered the ethical implications of our work and have strived to conduct our research with the highest standards of scientific integrity and responsibility. We outline the specific considerations below.

1. **Broader Impact and Potential for Harm**

   This research aims to build more capable AI agents for simulated environments, such as games and robotics, by enhancing their long-term memory. While video generation technologies pose a dual-use risk (e.g., deepfakes), our model's application is confined to the non-photorealistic, domain-specific world of Minecraft. This focus significantly mitigates the potential for misuse in creating malicious real-world synthetic media.

2. **Data and Privacy**

   Our research utilizes established public datasets (VPT, WorldMem) and a new benchmark we generated using the MineDojo simulator. All data consists of anonymized Minecraft gameplay and contains no Personally Identifiable Information (PII). To promote reproducibility and further research, we commit to open-sourcing our generated dataset upon the paper's acceptance.

## 7 REPRODUCIBILITY STATEMENT

To ensure the reproducibility of our results, we provide comprehensive details of our methodology, experimental setup, and resources. Our core framework, Memory Forcing, is described in Section 3, with specific architectural details in Section 3.2 and 3.4, and our Memory forcing training strategy in Section 3.3. The complete experimental setup, including implementation details, hardware (NVIDIA H100/H20 GPUs), and key hyperparameters, is detailed in Section 4.1. The datasets used, including public benchmarks (VPT, WorldMem) and our newly generated evaluation data, are also described in Section 4.1. We provide a full breakdown of our evaluation metrics and comparisons against baselines in Section 4.2. Furthermore, we commit to releasing our source code and the newly generated dataset upon acceptance of this paper.

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

## A APPENDIX

### A.1 DECLARATION OF LLM USAGE

Large Language Models (LLMs) were used as general-purpose assistive tools to improve the grammar and clarity of this manuscript. The core scientific contributions, including the methodology, experimental design, and analysis, are entirely our own. The authors have reviewed all text and take full responsibility for the content of this paper.

### A.2 CHAINED FORWARD TRAINING ALGORITHM

---

**Algorithm 1** Chained Forward Training (CFT)

---

**Require:** Video $\mathbf{x}$, conditioning inputs $\mathcal{C}$, forward steps $T$, window size $W$, model $\epsilon_\theta$

1: Initialize $\mathcal{F}_{\text{pred}} \leftarrow \emptyset$, $\mathcal{L}_{\text{total}} \leftarrow 0$
2: **for** $j = 0$ to $T - 1$ **do**
3:  Construct window $\mathcal{W}_j[k] \leftarrow \mathcal{F}_{\text{pred}}[k]$ if $k \in \mathcal{F}_{\text{pred}}$ else $\mathbf{x}_k$ for $k \in [j, j + W - 1]$
4:  Compute $\mathcal{L}_j \leftarrow \|\epsilon - \epsilon_\theta(\mathcal{W}_j, \mathcal{C}_j, t)\|^2$, update $\mathcal{L}_{\text{total}} \leftarrow \mathcal{L}_{\text{total}} + \mathcal{L}_j$
5:  If $j < T - 1$: $\mathcal{F}_{\text{pred}}[j + W - 1] \leftarrow \hat{\mathbf{x}}_{j+W-1}$ {Fewer denoising steps}
6: **end for**
7: **return** $\mathcal{L}_{\text{chain}} \leftarrow \mathcal{L}_{\text{total}}/T$

---

### A.3 DATASET DETAILS

We utilize the WorldMem (Xiao et al., 2025) dataset together with three additional datasets collected using MineDojo (Fan et al., 2022) to evaluate our model across multiple dimensions: memory capacity, generalization abilities, scene exploration, and efficiency. The WorldMem (Xiao et al., 2025) dataset contains 150 video sequences of 1,500 frames each, sampled from five terrain types: *ice_plains*, *desert*, *savanna*, *sunflower_plains*, and *plains*. The Scene Generation dataset includes 300 sequences of 800 frames each, while the Efficiency dataset consists of 20 sequences of 4,000 frames. The Generalization Abilities dataset comprises 150 sequences of 800 frames each, sampled from nine unseen terrains: *extreme_hills*, *taiga*, *stone_beach*, *swampland*, *river*, *beach*, *mesa*, *frozen_ocean*, and *forest_hills*. Dataset statistics are summarized in Table 4.

For action configurations, all MineDojo-collected datasets adopt the same setup as WorldMem, including movement actions (*left*, *right*, *forward*, *back*) and view-control actions (*look_up*, *look_down*, *turn_left*, *turn_right*). The Memory Capabilities dataset constrains agents within confined regions with diverse actions including vertical movement. For Scene Generation and Generalization datasets, we use a two-phase strategy: initial 600 frames with full action diversity, followed by restricted actions (*forward*, *turn_left*, *turn_right*) to assess generation capabilities.

### A.4 MODEL COMPARISON

Table 5 provides a comprehensive comparison of our approach with existing video generation models for Minecraft environments, highlighting key differences in memory mechanisms, storage efficiency, and capabilities across different methodological paradigms.

Table 4: Dataset statistics for model evaluation.

| Dataset | Video Count | Frames per Video | Terrain Types |
|---|---|---|---|
| Efficiency | 20 | 4,000 | 5 |
| Generation Performance | 300 | 800 | 5 |
| Generalization Performance | 150 | 800 | 9 |
| Long-term Memory | 150 | 1,500 | 5 |

Table 5: Comparison of Video Generation Models on Minecraft

| Method | Memory Type (Complexity) | Memory Storage | Memory Scope | Dataset | Size | Video Length | Actions |
|---|---|---|---|---|---|---|---|
| Oasis | / | / | / | VPT | 2.5M | 64 frames | 25 |
| Mineworld | / | / | / | VPT | 2.5M | 64 frames | 25 |
| NFD | / | / | / | VPT | 2.5M | 64 frames | 25 |
| LSVM | State Space Model (O(1)) | Compressed hidden state | Limited by training length | TECO (Yan et al., 2023) | 200K | 150 frames | 4 |
| VRAG | Similarity-based RAG (O(1)) | Fixed-length buffer | Limited by buffer size | MineRL (Guss et al., 2019) | 1K | 1200 frames | 5 |
| Worldmem | Pose-based RAG (O(n)) | Memory bank (Stores all frames) | Long-term | MineDojo | 20K | 1500 frames | 8 |
| **Ours** | **3D-based RAG (O(1))** | **Scene-dependent sparse storage** | **Long-term** | **VPT+MineDojo** | **85K** | **1500 frames** | **25** |

The first group represents traditional autoregressive video models without explicit memory mechanisms. Models like Oasis (Decart et al., 2024), Mineworld (Guo et al., 2025), and NFD (Cheng et al., 2025) rely solely on temporal context windows and demonstrate strong performance in scene generation but suffer from spatial inconsistency when revisiting previously encountered areas due to their limited memory scope.

The second group encompasses recent memory-augmented approaches that attempt to extend model capabilities through various memory mechanisms. LSVM (Po et al., 2025) employs state space models to compress historical information into hidden states, achieving constant-time complexity but with memory scope fundamentally limited by training sequence length. VRAG (Chen et al., 2025) utilizes similarity-based retrieval with fixed-length buffers, providing constant-time access but constraining long-term memory capacity. WorldMem implements pose-based retrieval with comprehensive frame storage, enabling true long-term memory but suffering from linear complexity growth as memory banks accumulate redundant information during extended sequences.

Our Memory Forcing framework uniquely combines the advantages of both paradigms while addressing their respective limitations. Unlike traditional models, we maintain long-term spatial memory through explicit 3D scene representation. Unlike existing memory-augmented approaches, our 3D-based retrieval system achieves constant-time complexity with scene-dependent sparse storage that adapts to spatial redundancy patterns. This design enables efficient scaling to extended sequences while preserving both spatial consistency and scene generation capabilities across the most comprehensive action space among compared methods.

## A.5 LIMITATIONS AND FUTURE WORK.

**Limitations.** While Memory Forcing demonstrates strong performance in memory retention and generation quality, several limitations remain. Our current implementation is primarily validated on Minecraft gameplay scenarios, which may not directly generalize to other environments without domain-specific adaptation. Additionally, our model operates at a fixed resolution of 384 × 224 pixels, which may limit visual detail in applications requiring higher fidelity.

**Future Work.** Future research should focus on extending our framework to diverse gaming environments and real-world scenarios at higher resolutions. We plan to explore domain adaptation techniques that preserve core memory mechanisms while accommodating different visual characteristics. Additionally, investigating simplified architectural designs that maintain memory advantages while reducing implementation complexity could enhance broader applicability. Integration with advanced acceleration techniques may further improve both efficiency and performance across diverse interactive scenarios.

### A.6 ADDITIONAL QUALITATIVE COMPARISONS

We present additional qualitative analyses of different models' performance in novel scene generation. Across Figures 6–8, our method shows superior spatial coherence, temporal continuity, and scene detail compared to baseline models in both familiar and unfamiliar terrains. Figure 5 demonstrates generalization on frozen ocean terrain. While WorldMem reproduces familiar training terrains like plains, our model successfully maintains the target frozen ocean environment, showing better generalization capabilities. Figures 6 and 7 compare performance across extreme hills, ice plains, and desert terrains. Baseline methods (Oasis, NFD, WorldMem) often generate unrealistic views, violate spatial consistency, or fail to reflect agent motion. Our approach maintains geometric and temporal coherence while producing high-quality novel scenes. Figure 8 examines long-term memory scenarios. Specialized long-term memory models struggle with novel scene generation and show limited generalization in new environments. Our model effectively uses long-term memory to generate consistent, realistic scenes while preserving spatial and temporal coherence. These comparisons demonstrate that our geometry-indexed spatial memory and generative approach delivers robust performance across diverse terrains, generalization tasks, and long-term memory scenarios, outperforming existing baselines.

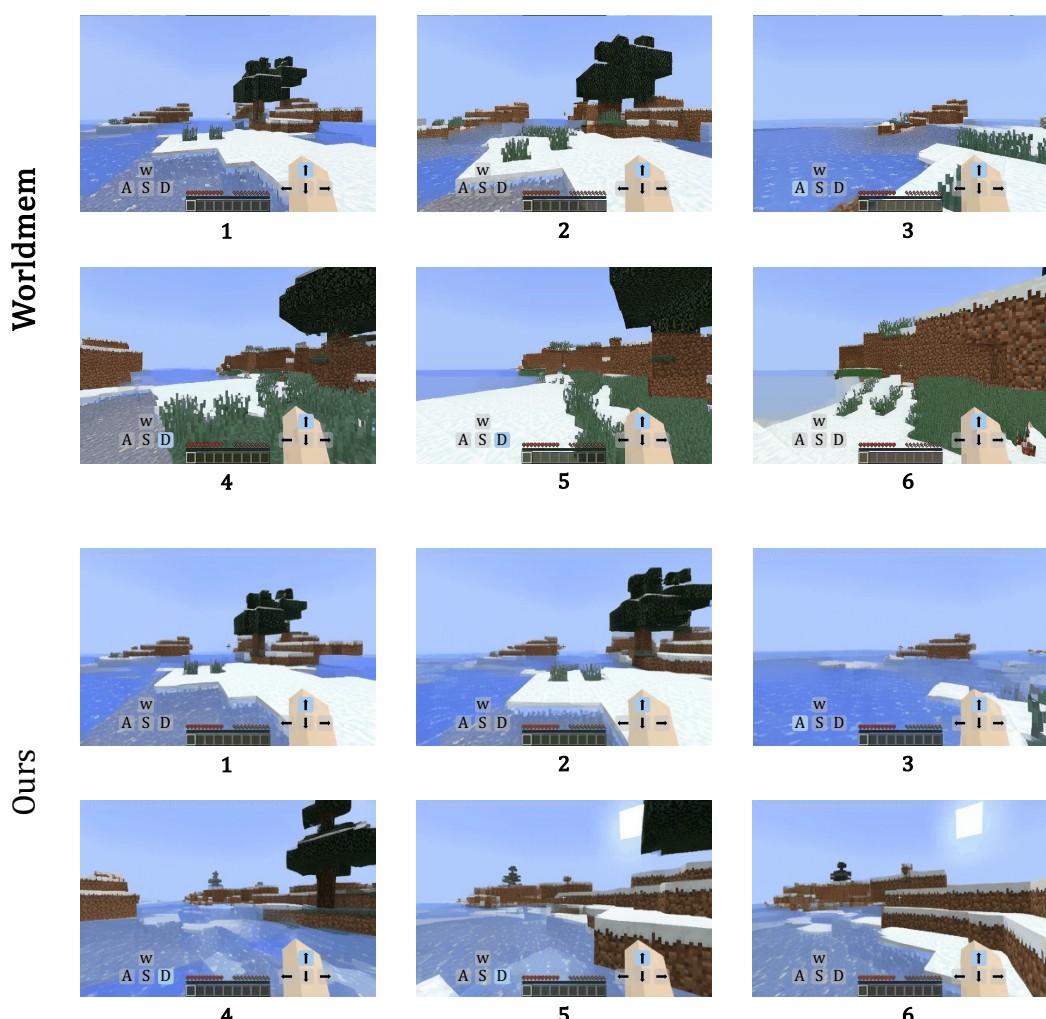

Figure 5: Generalization performance on frozen ocean. When generating frozen ocean terrain, WorldMem (Xiao et al., 2025) produces novel scenes resembling the plains terrain from the training set. By contrast, our model preserves the frozen ocean terrain across novel scene generations.

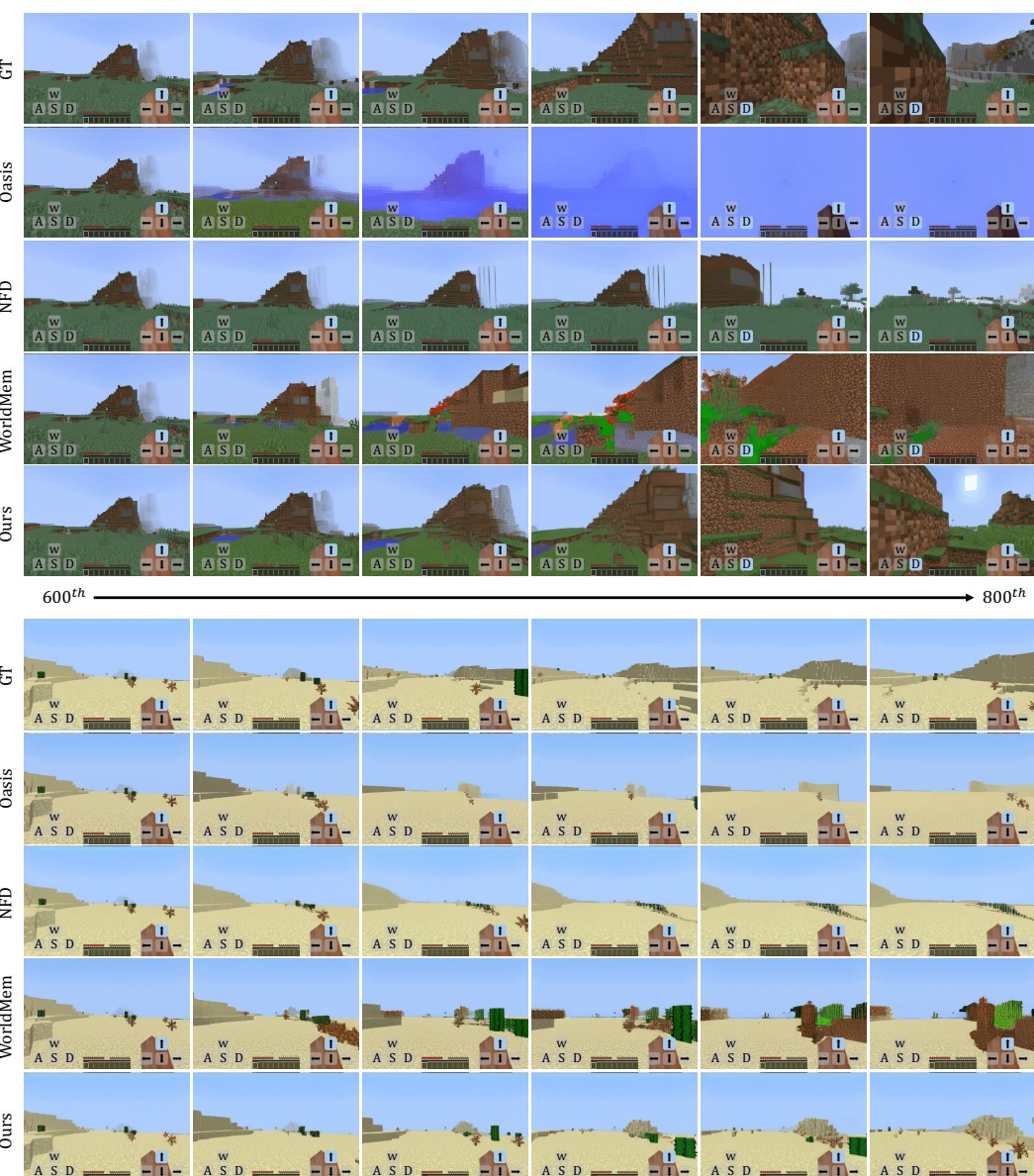

Figure 6: Qualitative results. Comparison of different models' novel scene generation on two terrains: extreme hills (top) and desert (bottom). In extreme hills, our method generates novel views while preserving spatial consistency, whereas Oasis (Decart et al., 2024) fails, collapsing to blue images. WorldMem (Xiao et al., 2025) and NFD (Cheng et al., 2025) produce unrealistic views that break spatial consistency. In desert, Oasis (Decart et al., 2024) and NFD (Cheng et al., 2025) fail to reflect the agent's forward motion, and WorldMem (Xiao et al., 2025) lacks temporal and spatial consistency. By contrast, our method maintains spatial coherence and produces rich, realistic novel views.

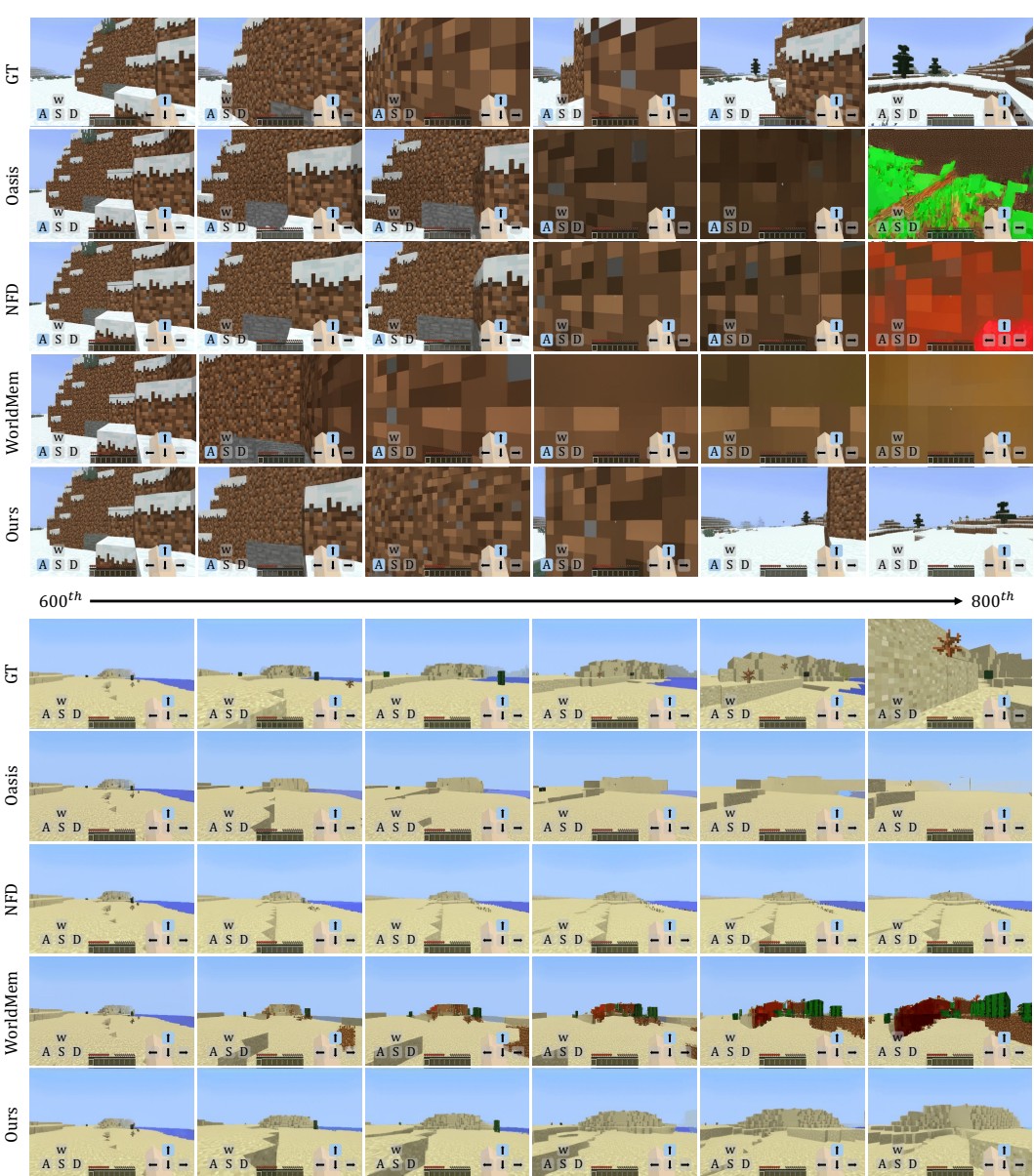

Figure 7: Qualitative results. Comparison of different models' novel scene generation on two terrains: ice plains (top) and desert (bottom). In the ice plains scenario, an action sequence drives the agent into a confined area, testing the models' memory in generating novel scenes. Oasis (Decart et al., 2024), NFD (Cheng et al., 2025), and WorldMem (Xiao et al., 2025) fail to produce correct views when the agent turns left and moves forward, remaining trapped. By contrast, our model successfully generates novel views after escaping while preserving the ice plains terrain. In the desert scenario, NFD (Cheng et al., 2025) fails to reflect the agent's forward motion, while World-Mem (Xiao et al., 2025) and Oasis (Decart et al., 2024) violate temporal and spatial consistency. Our method consistently maintains spatial coherence and generates realistic novel views.

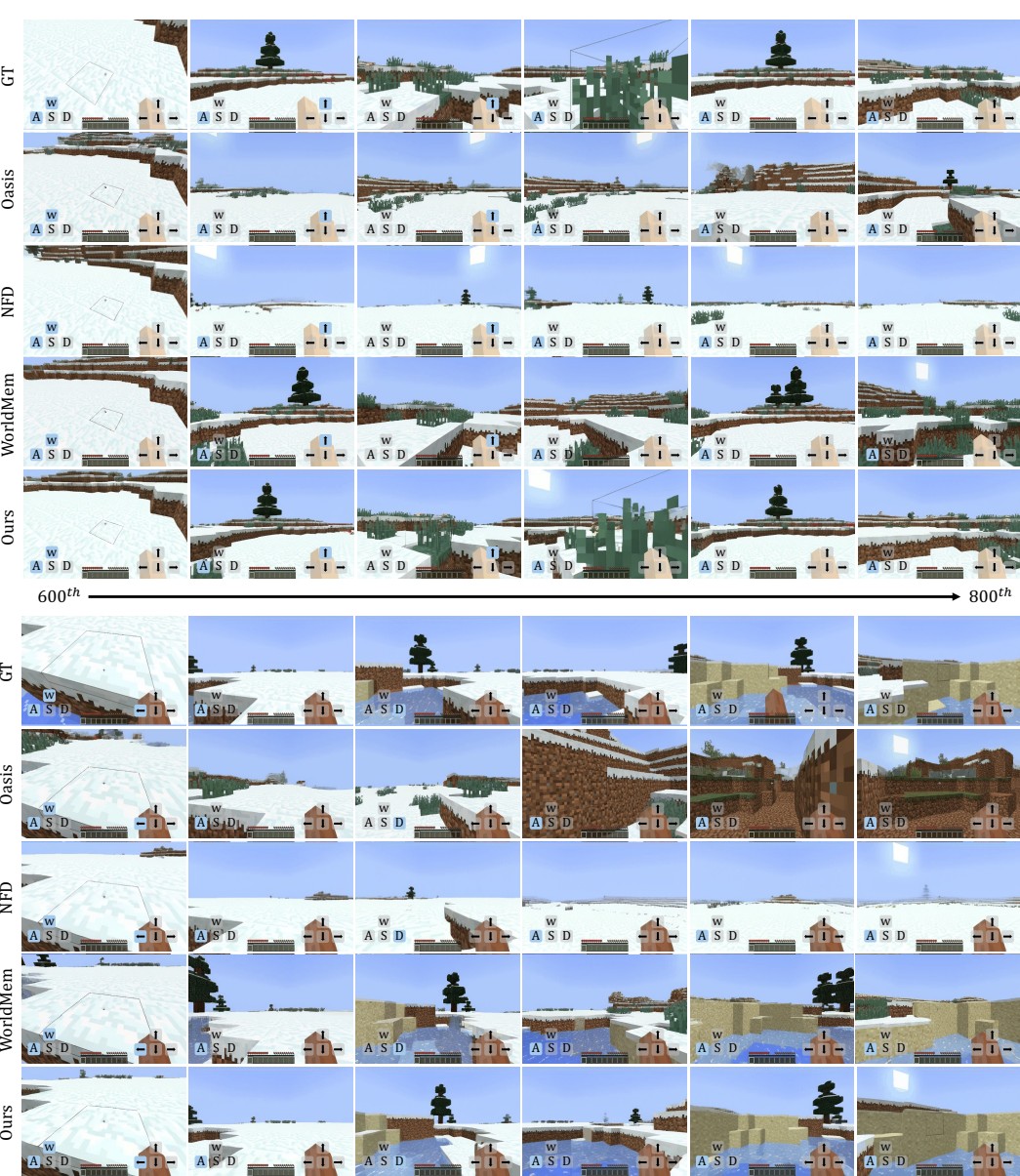

Figure 8: Qualitative results on long-term memory across different models. We compare the generative capabilities of different models under long-term memory settings. Our model achieves the best spatial consistency, temporal continuity, and preserves rich scene details.

