# OpenReview forum: "Memory Forcing: Spatio-Temporal Memory for Consistent Scene Generation on Minecraft"
_ICLR.cc/2026/Conference — ICLR 2026 Conference Withdrawn Submission_

### Official Review · Reviewer_wc9N · 2025-10-30

**Soundness:** 1
**Presentation:** 2
**Contribution:** 1
**Rating:** 2
**Confidence:** 5

**Summary:**

The method in this paper combines Long-term Spatial Memory and Short-term Temporal Memory context learning to achieve long-term consistent video generation in Minecraft.

**Strengths:**

The method proposed in this paper shows advantages in the quantitative metrics provided by the authors.

**Weaknesses:**

- (1) The most confusing point in this paper is: Is there enough evidence to show that Long-term Spatial Memory leads to poorer quality when generating new scenes? The authors seem to take this as a given fact, which is very hard to believe. Could you provide more experimental results to support this claim? If this cannot be proven, then the method proposed in this paper would be nothing but baseless story and would lack any value. Furthermore, I looked at the results from Context-as-Memory, which seems to use Long-term Spatial Memory, yet it also performs very well in generating new scenes. Is it possible that with sufficient training, Context Learning could simultaneously handle revisiting old scenes and generating new ones? Is it really necessary to distinguish between Long-term and Short-term memory?
- (2) Even without considering the correctness of the motivation behind "Long-term Spatial Memory leads to poorer quality when generating new scenes," what is the fundamental difference between the method proposed in this paper, which uses both Short-term and Long-term contexts as conditioning for learning, and previous context learning methods such as WorldMem, Context-as-Memory, and VMem? The Context-as-Memory method, which is based on FOV retrieval, is highly likely to retrieve long-term context as well as short-term context. Similarly, VMem's approach, based on point cloud reconstruction matching, also follows this idea. The GEOMETRY-INDEXED SPATIAL MEMORY proposed by the authors seems very similar to the VMem approach. Overall, no matter how the authors present their method, when looking at how the method actually works, it is very similar to existing approaches. This raises doubts about the novelty and contributions of the paper.
- (3) Although the quantitative metrics proposed in this paper seem to show advantages, the qualitative results presented do not clearly or directly reflect the method's capabilities in terms of memory. It appears that several figures only show a few key frames from a small segment of long videos, without trajectory hints or revisit annotations. This makes it difficult for readers to clearly perceive the advantages of the proposed method in terms of memory. This further intensifies the doubts regarding the true effectiveness of the method.

**Questions:**

none

---

### Official Review · Reviewer_5aog · 2025-10-30

**Soundness:** 2
**Presentation:** 2
**Contribution:** 1
**Rating:** 2
**Confidence:** 4

**Summary:**

The proposes a training framework for autoregressive video diffusion models for spatial consistency and generative quality. It introduces Hybrid Training and Chained Forward Training to teach models when to rely on temporal vs. spatial memory, together with a Geometry-indexed Spatial Memory built via streaming 3D reconstruction and point-to-frame retrieval.

**Strengths:**

1. The paper’s writing is generally clear, though some sections could benefit from more concise explanations.

2. The Figure 1 and 2 are reasonably clear and informative, but certain visual details could be better highlighted for clarity.

**Weaknesses:**

### Major Weakness

**1. Risk of Overclaim**: The paper tends to overclaim its contribution by repeatedly using the term *“world modeling”* in the paper, but its contexts are more aligned with *gaming video generation*. Its use of *“interactive game world modeling”* blurs the established common sense of world models in the community, which typically take the current world state (observation) and learn to predict future states after a (series of) actions or uncertainties [1], and then used to interact with the real world in downstream tasks [2]. Although similar concept mentioned in line 170-171 (which is basically all AR and AR-diffusion video models do), the paper only uses Minecraft synthetic data to verify spatial memory and geometry consistency, which would be more significant and applicable in real-world scenarios.

**2. Missing important baselines:** The main architectural contributions (Spatial Memory Augmentation and Training Strategy) of the paper are developed on top of the AR diffusion backbone. However, since current video foundation models are already well developed, the paper does not compare or conduct any ablations against other open-sourced AR and AR-diffusion video models and training methods such as Self-Forcing [3], CausVid [4], and MAGI-1 [5], etc.

**3. Lack of motivation on gaming video generation:** Although the paper mentions the challenges of gaming videos in temporal and spatial memory, it does not ground its motivation in why it chooses the paradigm of gaming video generation instead of game agents in simulations such as Voyager [6]. As the authors repeatedly claim world modeling, I have to mention that simulations can more properly capture physical constraints (such as gravity and velocity) [7]. Despite the paper only discussing geometry, physical plausibility cannot be ignored in world modeling [8].

**4. Accumulation error and artifacts in visualization:** The paper showcases some demos demonstrating long-term and spatio-temporal memory. However, it is evident that there is significant noise and shape degradation in the form of accumulation errors (especially on objects such as brown land and green trees), which are likely caused by the AR diffusion models. These artifacts significantly affect the overall quality of the generated videos. Nevertheless, the paper does not concretely discuss or propose any method to address the accumulation error.

### Minor Weakness

1. In the main paper, Figures 3 and 4 present several visualizations and occupy almost two pages, but they still fail to clearly convey the model’s advantages and the authors’ intended message. Especially in such a compact layout (9-page single column) under the ICLR template.

**Reference**

[1] Videoworld: Exploring knowledge learning from unlabeled videos (CVPR 2025)

[2] Structured World Models from Human Videos (RSS 2023)

[3] Self Forcing: Bridging the Train-Test Gap in Autoregressive Video Diffusion (NeurIPS 2025 Spotlight)

[4] From Slow Bidirectional to Fast Autoregressive Video Diffusion Models (CVPR 2025)

[5] MAGI-1: Autoregressive Video Generation at Scale

[6] Voyager: An Open-Ended Embodied Agent with Large Language Models

[7] NVIDIA Isaac Sim

[8] VLIPP: Towards Physically Plausible Video Generation with Vision and Language Informed Physical Prior (ICCV 2025)

**Questions:**

Please see the details in the weakness section, and I hope the authors can address my concerns from the following perspectives.

1. Would you provide a more comprehensive evaluation not only on gaming videos but also on real-world data, since both geometric consistency and world modeling claimed in the paper rely on these evaluations?

2. Include a more comprehensive and fair set of baselines.

3. Accumulation errors in visualization, and why long-term memory enhancement in the paper fails to address these artifacts.

---

### Official Review · Reviewer_pkTY · 2025-11-03

**Soundness:** 3
**Presentation:** 3
**Contribution:** 3
**Rating:** 6
**Confidence:** 3

**Summary:**

The paper introduces the Memory Forcing framework to resolve the fundamental trade-off between long-term spatial consistency and novel scene generation quality in autoregressive video diffusion models for open-world environments like Minecraft. This paper uses incremental 3D reconstruction and a Point-to-Frame Retrieval mechanism with $O(1)$ complexity, achieving 7.3x faster retrieval and 98.2% less memory usage than baselines for long sequences. A novel training protocol is proposed that adaptively uses temporal memory for exploration and spatial memory for revisiting areas, ensuring both flexibility and consistency. Memory Forcing significantly outperforms state-of-the-art baselines across metrics for long-term memory, generalization, and generation quality, successfully balancing consistency with visual realism.

**Strengths:**

1. Successfully balances long-term spatial consistency (when revisiting areas) with high-quality novel scene generation.
2. Introduces Geometric-Indexed Spatial Memory with $O(1)$ complexity Point-to-Frame Retrieval.
3. Uses Hybrid Training to adaptively utilize temporal vs. spatial memory and Chained Forward Training (CFT) to ensure robust geometric consistency during pose changes.
4. Outperforms state-of-the-art models across all key metrics (FVD, PSNR, LPIPS) in long-term memory and generalization tasks.

**Weaknesses:**

1. The current implementation is primarily validated on the Minecraft domain. I think the materials and style of the Minecraft virtual scenes are indeed quite limited. Long-term memory may also not be suitable for most real-world scenarios.
2. The graphics quality of Minecraft is essentially that of low-dimensional vector-based visuals.

**Questions:**

1. This paper strikes me as being very similar to topics studied in the SLAM field. The authors may need to conduct some research and discussion regarding this.
2. Based on the demonstration, the scenes seem quite simple, and it's difficult to see where a long-term memory mechanism would be specifically required.

---

### Official Review · Reviewer_ipK7 · 2025-11-07

**Soundness:** 3
**Presentation:** 3
**Contribution:** 3
**Rating:** 6
**Confidence:** 2

**Summary:**

This paper presents memory forcing, a spatio-temporal memory for consistent scene generation in interactive world modeling. IN particular, memory forcing is a learning framework that pairs training protocols with a geometry-indexed spatial memory. Hybrid training covers both revisiting explored areas and exploration of new areas. Other details, like chained forward training and point-to-frame retrieval, are also introduced to maintain the spatio-temporal consistency for video generation. The proposed method has been validated on Minecraft videos. Experimental results demonstrate the effectiveness of the proposed method.

**Strengths:**

1.[effectiveness] The quantitative results of the proposed method is clearly outperforming the previous methods. The qualitative results are also telling the same story. For instance, in the last row and last column of figure 3, the shape of tree is better memorized than the previous method. That explains a clearly lower FVD compared to the previous method.

2.[efficiency] The proposed method is much faster than the previous method as indicated in table 2.

**Weaknesses:**

1.[ablation] Why does the point-to-frame retrieval use a fixed number of top-8 frames (L252)? What is the implication if we choose a different number such as top-1 or top-100?

2.[clarity] I think the point-to-frame retrieval is the key element in the proposed method for such a good performance. If we increase the number of retrieved frames, how does it impact the speed and memory as shown in Table 2?

**Questions:**

See the weaknesses.

---

### Note · Authors · 2025-11-14

I have read and agree with the venue's withdrawal policy on behalf of myself and my co-authors.